# A Panorama of Immune Fighters Armored with CARs in Acute Myeloid Leukemia

**DOI:** 10.3390/cancers15113054

**Published:** 2023-06-05

**Authors:** Ilias Christodoulou, Elena E. Solomou

**Affiliations:** 1Department of Internal Medicine, University of Pittsburgh, Pittsburgh, PA 15213, USA; christodouloui@upmc.edu; 2Department of Internal Medicine, University of Patras Medical School, 26500 Rion, Greece

**Keywords:** acute myeloid leukemia, chimeric antigen receptors, CAR-T cells, CAR-NK cells, adoptive cell therapy

## Abstract

**Simple Summary:**

Acute myeloid leukemia (AML) is a difficult-to-treat cancer, and chemotherapy can cause severe side effects. Patients may need stem cell transplantation, but this is challenging and not always successful. Even with intensive treatment, some patients may relapse or develop a refractory disease. Genetically modifying some white blood cells, called lymphocytes, with specific molecules called chimeric antigen receptors (CARs) can generate new therapies that can target AML. In this study, we gather all current data from the literature regarding the clinical testing of CAR-based therapies against AML. Moreover, we analyze the limitations of the most established therapy, CAR-T cells (T-lymphocytes with CARs). Additionally, we provide some insights into the potential benefits of using alternative lymphocytes, such as the CAR-NK cells (natural killer cells with CARs).

**Abstract:**

Acute myeloid leukemia (AML) is a devastating disease. Intensive chemotherapy is the mainstay of treatment but results in debilitating toxicities. Moreover, many treated patients will eventually require hematopoietic stem cell transplantation (HSCT) for disease control, which is the only potentially curative but challenging option. Ultimately, a subset of patients will relapse or have refractory disease, posing a huge challenge to further therapeutic decisions. Targeted immunotherapies hold promise for relapsed/refractory (r/r) malignancies by directing the immune system against cancer. Chimeric antigen receptors (CARs) are important components of targeted immunotherapy. Indeed, CAR-T cells have achieved unprecedented success against r/r CD19+ malignancies. However, CAR-T cells have only achieved modest outcomes in clinical studies on r/r AML. Natural killer (NK) cells have innate anti-AML functionality and can be engineered with CARs to improve their antitumor response. CAR-NKs are associated with lower toxicities than CAR-T cells; however, their clinical efficacy against AML has not been extensively investigated. In this review, we cite the results from clinical studies of CAR-T cells in AML and describe their limitations and safety concerns. Moreover, we depict the clinical and preclinical landscape of CAR used in alternative immune cell platforms with a specific focus on CAR-NKs, providing insight into the future optimization of AML.

## 1. Introduction

Acute myeloid leukemia (AML) is a malignancy of bone marrow (BM) characterized by differentiation arrest and the uncontrolled proliferation of myeloid progenitors, known as blasts [1]. AML is the most common form of leukemia in adults and the second most common in children, with an overall 5-year survival rate of 30.5% [2]. Intensive chemotherapeutic regimens still make up the backbone and frontline of treatment protocols for the induction and maintenance of remission [3]. However, cytotoxic therapy is associated with severe toxicities, and a significant subset of patients are refractory to treatment or eventually relapse [4]. Refractory disease is the failure to achieve complete remission (CR; defined as BM blasts < 5%, absence of peripheral blasts, and extramedullary disease) or CR with incomplete hematologic recovery (CRi; defined as CR with residual cytopenias) after two cycles of induction therapy [3]. On the other hand, relapsed disease is defined as the reappearance of blasts in BM (>5%), peripheral blood (PB), or extramedullary sites in patients who had previously achieved CR [3]. The prognosis of relapsed/refractory (r/r) AML is extremely poor with an overall reported 5-year survival rate of less than 15% [5]. To date, allogeneic hematopoietic stem cell transplantation (allo-HSCT) is the only potentially curative option in high-risk AML, including r/r AML. However, 30–40% of patients treated with HSCT relapse, depriving clinicians from further options [6,7]. Moreover, allo-HSCT is typically reserved for patients with low disease burden since high disease burden is associated with an increased risk of graft failure and relapse. Therefore, patients with chemorefractory disease require additional therapies to decrease the disease burden before allo-HSCT. Targeted therapies are being investigated as primary or adjunct therapies to achieve and maintain remission in high-risk AML [8].

Chimeric antigen receptors (CARs) are engineered receptors that can selectively target specific antigens and are composed of different serially fused molecular domains [9]. More specifically, the CAR’s extracellular moiety, the single-chain variable fragment, recognizes and binds to the antigen of interest; the hinge and transmembrane domains serve with antigen-binding flexibility and cell-membrane anchoring, respectively, while the combination of intracellular domains is responsible for signal transmission and the triggering of a specific cellular function (Figure 1) [10]. The genetic modification of immune cells for CAR expression on their surface has been used in the context of targeted immune therapies, primarily on T-cells, NK cells, and other immune effectors [11]. CAR-T cells have achieved unprecedented success in CD19+ malignancies, which can be attributed in a great extent to antigen selection, as CD19 is expressed at high levels on malignant B lymphocytes with concurrent confined expression only in normal B cells and no other healthy tissues [12]. Therefore, it is evident that CD19-CAR-T cells can eliminate cancer cells by targeting “on-tumor” CD19 and concurrently result in normal B-cell aplasia (“on-target/off-tumor effect”), which can be utilized as a tool for measuring CAR-T cell persistence after desirable outcomes have been achieved [13,14,15]. The major CD19-specific adverse event is the resultant hypogammaglobulinemia, which can be tackled with exogenous immunoglobulin administration [16,17]. On the other hand, antigen selection has been the most challenging aspect of CAR-based immune therapies in AML, with several antigens already tested in the clinical setting. However, thus far, none of them satisfy the ideal combination of universal expression in all AML cases, high levels on the surface of AML blasts and leukemic stem cells (responsible for self-renewal of the cancer), and a lack of expression in normal hematopoietic or endothelial tissues, with ongoing efforts to identify more ideal antigen targets [18].

CAR-T cells have been widely investigated in several trials of AML, and important conclusions have been drawn regarding their outcomes, safety, and limitations. Other effector immune cells, such as CAR-engineered NK cells, have been extensively studied in preclinical models of AML, and while they hold promise and unique advantages over CAR-T cell therapy, their bedside application has been limited so far. In this review, we discuss the current data on CAR-based immunotherapies in AML, with particular emphasis on clinical studies of CAR-T cells and preclinical studies of CAR-NK cells and other effectors. Moreover, we examine the limitations of each cellular immunotherapy and propose solutions to increase the anti-AML efficacy of CAR-NK therapy concurrently by taking advantage of the potential benefits of CAR-NK treatment, especially in the safety profile.

## 2. CAR-T Cells in AML

### 2.1. Clinical Testing of CAR-T Cells in AML

Current clinical studies on CAR-T cells in AML have primarily focused on patients with r/r disease, with the ultimate goal of bridging to HSCT (Table 1). The first clinical study of CAR-T cells in AML was published by Ritchie et al., and targeted the Lewis Y antigen, a fucosylated oligosaccharide [19]. Although the study showed trafficking of CAR-T cells at sites of AML infiltration and persistence of CAR-T cells in vivo for up to 10 months without serious acute toxicities, there were no clinical responses, and all initial responders eventually experienced disease progression [19]. CD123, the α-chain of IL3R, is a cell surface protein that is overexpressed in various hematological malignancies, including AML blasts and LSCs [20]. Autologous anti-CD123 CAR-T cells have been used in several studies, with short-lived clinical responses [21,22,23]. These studies have generally demonstrated a low rate of serious adverse events and a low persistence of CAR-T cells. Another study explored the effect of adding alemtuzumab, an anti-CD52 monoclonal antibody, to the lymphodepleting regimen (LD) before infusing an allogeneic anti-CD123 CAR-T cell product with disrupted TRAC (to minimize graft-versus-host disease risk) and CD52. The use of alemtuzumab in the LD led to a greater expansion of anti-CD123 CAR-T cells, but there were no differences in clinical efficacy compared to the control LD group [24].

CLL-1 (also known as CLEC12A) is a transmembrane glycoprotein that is highly expressed in acute myeloid leukemia (AML) blasts, leukemia stem cells (LSCs), and monocytes but has low expression in normal hematopoietic cells [25]. This makes it a promising target for chimeric antigen receptor (CAR) T-cell therapy. In a study by Zhang et al., CLL1-CAR T cells were investigated in four pediatric patients with r/r AML, resulting in CR in three out of four patients. However, all patients eventually succumbed to disease or HSCT toxicity within 24 months [26]. In a follow-up phase I/II trial, the same research group recruited eight pediatric patients with r/r AML, and six out of eight patients achieved CR, with four out of six having a morphologic leukemia-free state (MLFS) or minimal residual disease (MRD) negativity. Eventually, five out of the six patients underwent HSCT, and four of them remained disease-free until the end of the study. All patients experienced grade 1–2 cytokine release syndrome (CRS); however, no immune effector cell-associated neurotoxicity syndrome (ICANS) or other major organ adverse events were reported [27]. In another preliminary report, CLL1-CAR T cells showed durable CR in two of three patients with r/r AML before HSCT was performed [28]. A study in adult patients with r/r AML also showed promising results, with CR achieved in six of ten patients (three with MRD negativity and three with MRD positivity). Four of the six patients underwent HSCT and remained alive and MRD-negative. One patient remained in the CR/MRD-negative state without transplantation, while another patient died due to severe agranulocytosis despite being MRD-negative. All treated patients developed severe pancytopenia and CRS (high-grade in six of ten patients), but none reported neurotoxicity [29]. Overall, CLL1-CAR T-cell studies have shown similar patterns of adverse events and meaningful responses, with approximately half of the patients being bridged to HSCT.

CD33 is a protein expressed in 85–90% of AML blasts and LSCs, as well as in normal myelopoiesis, specifically in early multilineage hematopoietic progenitor and more mature myeloid cells, but not in stem cells or multipotent progenitor cells [30]. Due to the high frequency and levels of CD33 expression, it has been proposed as a target for CAR-T cell therapy. An early clinical trial with autologous CD33-CAR T cells in patients with refractory AML did not show significant clinical responses, although CAR-T cell activity was noted in two of three patients, as evidenced by cytokine release syndrome (CRS) [31]. Similarly, CD33-CAR T cells in an adult patient with refractory AML induced a transient decrease in BM blasts, but the disease eventually progressed [32]. Further investigation focused on the dual targeting of CD33 and CLL1, as it has been reported that in approximately 67% of AML cases, blasts co-express CLL1 and CD33 [33]. Liu et al., infused r/r AML patients with a CLL1-CD33 compound CAR (cCAR)-T cell product after fludarabine/cyclophosphamide pre-conditioning. Their preliminary data showed that eight out of ten patients achieved MRD-negative CR, with seven of them eventually receiving HSCT, which was the primary objective of the study [34]. However, high-grade pancytopenia was observed in all patients, while CRS of varying degrees was noted in nine of the ten patients, and ICANS in five of the ten patients.

CD7 is a transmembrane glycoprotein that is highly expressed in approximately one-third of AML cases but not in normal myeloid cells [35]. T cells express CD7, which may result in a fratricide effect if CAR-T cells targeting CD7 are infused into the patients [36]. Hu et al., conducted a study on CD7KO CD7-CAR-T in patients with CD7-positive hematologic malignancies, including one patient with CD7-positive r/r AML who achieved CRi and underwent allo-HSCT three months after CAR-T infusion [37]. Another approach is to target MICA/MICB antigens on the surface of AML cells using NKG2D-CAR-T cells. However, two preliminary studies and one completed clinical study have shown a lack of substantial clinical efficacy of NKG2D-CAR T cells in r/r AML, which was associated with poor in vivo persistence of CAR-T cells, possibly due to the fratricide effect of targeting MICA/MICB in CAR-T cells [38,39,40]. To overcome this issue, one study investigated the knockdown of MICA/MICB in NKG2D-CAR T cells, which resulted in a relative improvement in CAR-T cell engraftment [40]. 

The efficiency of CAR-T cells depends in a great extent on the persistence of these immune effectors in the body [41]. The effective detection of the CAR in peripheral blood, bone marrow, and CSF is pivotal for the monitoring of the clinical outcomes and toxicities of patients treated with CAR-T cells, especially in the setting of clinical trials. There are multiple ways to detect CAR-engineered T cells in the body. Genomic DNA can be isolated from peripheral blood cells and used as a template for the determination of the quantity of CAR-modified cells by calculating vector copy numbers with real-time quantitative polymerase chain reaction (qPCR) or digital PCR (dPCR) [42]. Moreover, CAR mRNA can be isolated and quantified with transcriptomic analysis, such as RNA sequencing, although it is less clinically popular than qPCR [42]. A good correlation of qPCR results has been shown for detection of the CAR-protein on the surface of T cells with flow cytometry [42,43,44]. In a comprehensive review of the literature from Turicek et al., the authors found that flow cytometry (used in around 43% of trials) and qPCR (used in around 57%) were the most commonly employed methods of CAR detection [45]. The authors also highlighted the need of universal approaches in detection techniques and time points as well as in the reporting of quantitative data [45]. While B-cell aplasia has been widely used to monitor the persistence of CD19-CAR-T cells, there is no such way to define “functional persistence” in AML, which necessitates the optimization of detection approaches for monitoring CAR-T cells in the blood.

**Table 1 cancers-15-03054-t001:** Completed CAR-T cell clinical studies with reported outcomes in relapsed/refractory acute myeloid leukemia. SD: Stable disease, PD: progressive disease, CR: complete response, MLFS: morphological leukemia-free state, N/A: information not available, KO: knock-out.

Target Antigen	Phase	CAR Construct/Genetic Modifications	Reported Outcomes
Lewis Y	I	[19] Ritchie et al.: CD28-CD3ζ	1/4 SD, 3/4 PD
CD123	I	[21] Cummins et al.: 4-1BB-CD3ζ	5/5 PD
	I	[22] Naik et al.: N/A	4/5 PD, 1/5 CR
	I	[23] Budde et al.: CD28-CD3ζ	1/6 MLFS, 1/6 CR, 4/6 PD or SD
CLL-1	I	[26] Zhang et al.: N/A	3/4 CR (eventually all succumbed to disease)
	I	[27] Zhang et al.: N/A	6/8 CR
	I	[28] Bu et al.: N/A	2/3 CR
	I	[29] Jin et al.: N/A	6/10 CR
CD33	I	[31] Tambaro et al.: 4-1BB-CD3ζ	3/3 PD
	I	[32] Wang et al.: 4-1BB-CD3ζ	1/1 PD
	I	[34] Liu et al.: compound CLL-1/CD33 CAR	8/10 CR
CD7	I	[37] Hu et al.: 4-1BB-CD3ζ/CD7KO	1/1 CR
MICA/MICB	I	[38] Baumeister et al.: NKG2D-CD3ζ	7/7 PD or SD

### 2.2. Limitations of CAR-T Cell Therapy in AML

#### 2.2.1. Toxicities and Manufacturing Hurdles

The effectiveness and feasibility of CAR-T cells for AML treatment are limited by various factors. One of the major limitations of any CAR-based cellular therapy for AML is the lack of AML-specific antigens, as the antigens expressed on AML are also present at various frequencies and densities in normal tissues, such as hematopoietic and endothelial tissues [46]. This issue is further complicated when using CAR-T cells because T cells persist longer in the body than other immune effectors. This can increase the likelihood of prolonged myeloablation after CAR-T cell infusion due to significant “on-target/off-tumor” toxicity [17]. Apart from protracted cytopenia, CAR-T cells have been associated with serious toxicities in clinical studies of AML, as mentioned above. CRS is a potentially life-threatening systemic inflammatory response that can occur following CAR-T cell infusion [17,47]. It is characterized by fever, hypotension, tachycardia, respiratory distress, and multiorgan dysfunction. CRS occurs due to the activation of CAR-T cells, which release cytokines such as IL-6, IFN-γ, and TNF-α, leading to the activation of macrophages. ICANS is a CAR-T-associated neurological toxicity, mediated by cytokine release and characterized by confusion, delirium, seizures, and cerebral edema [17,47]. CAR-associated hemophagocytic lymphohistiocytosis (carHLH) is a rare and potentially life-threatening toxicity that is mediated by cytokine release and is characterized by fever, cytopenia, hypertriglyceridemia, and high ferritin levels [48]. Another challenge is the manufacturing of CAR-T cells for AML, primarily because of the need for autologous CAR-T cells to prevent GVHD [49]. The process of collecting, modifying, and expanding a patient’s T cells can be time-consuming and complicated, especially in the case of aggressive cancers such as AML [50]. Additionally, autologous CAR-T cells are often influenced by the patient’s age, previous treatments (usually heavily pretreated), and disease stage (usually decided in r/r disease), which can affect the efficacy of the therapy. To address some of these challenges, allogeneic CAR-T cell products and strategies are being developed, including knockout of the T-cell receptor (TCR) or the engineering of gamma-delta (γδ) T cells [51,52]. However, a small percentage of TCR-positive T cells can persist after KO and cause GVHD, whereas certain γδ-T cell subsets can have a tumor facilitating effect [53,54]. Therefore, these strategies require further investigation before clinical application.

#### 2.2.2. Antigen Downregulation and Immune Suppression from AML

Target antigen loss or downregulation is a common mechanism employed by AML blasts to evade immune surveillance and is another key limitation of CAR-T cell therapy in AML. CAR-T cells rely exclusively on the recognition and binding of the CAR-antigen in order to exert their cytotoxic effects [55]. Therefore, potential antigen downregulation or loss (which is commonly seen with CD19 CAR) can render CAR-T cells ineffective unless multiple antigens are targeted, which is challenging in the limited antigenic repertoire of AML [56]. Moreover, AML cells can upregulate inhibitory ligands that bind to the checkpoint receptors in CAR-T cells, such as PD-1, TIM-3, CTLA-4, TIGIT, and GITR, and prevent their function by inducing exhaustion or anergy [57,58,59,60,61]. Another mechanism is the expression of immune-modulating enzymes from AML blasts, such as arginase II, indoleamine 2,3 dioxygenase, and ectonucleotidases, which lead to the accumulation of immunosuppressive metabolites, such as adenosine and kynurenine, and then the suppression of CAR-T cell proliferation and effector functions [46]. A similar mechanism of immune evasion involves AML cells that induce the accumulation of regulatory T cells (Tregs) and myeloid-derived suppressor cells (MDSCs) in the tumor microenvironment, which can inhibit CAR-T cell activity through direct cell-to-cell contact, metabolite secretion (including reactive oxygen species and nitric oxide), or cytokine-mediated suppression (including IL10 and TGF-β) [46]. Therefore, it is essential to develop novel strategies to overcome these immunosuppressive mechanisms and improve the efficacy of CAR-T cell therapy in AML. Combinational therapies of CAR-T cells with checkpoint inhibitors, small molecules that directly target immunosuppressive enzymes, or immune-modulating cell populations have been investigated in the preclinical setting of AML but have not yet been launched in a clinical study. 

All the major limitations of CAR-T cells in AML, that are discussed, are presented in Table 2.

## 3. Current State of CAR-NK Therapy in AML

### 3.1. Preclinical and Clinical Studies of CAR-NKs in AML

The challenges associated with CAR-T cell therapy in AML have shifted the focus to new approaches, such as CAR-NK cell therapy, which has gained momentum as a promising alternative. CAR-based immunotherapies for AML have used various sources of NK cells, including the NK cancer cell line NK92 and peripheral blood-derived NK cells. In the first completed clinical trial of CAR-NK cells for AML, NK92 MI cells engineered with CD33-CAR and constitutively secreting human IL2 were utilized in three patients with r/r AML co-managed with salvage chemotherapy (NCT02944162) [62]. Although no toxicity related to CAR-NK92 therapy was observed, no objective responses were achieved after three doses of CAR-NK92 cells at a maximum dose of 5 × 10^9^ cells [62]. Several preclinical studies have utilized NK92 cell lines engineered with various CARs. NK92 cells engineered with a CD276-CAR (with CD276 found to be expressed at high levels in AML cell lines) and further edited with CRISPR/Cas9 knockout of the inhibitory receptors CBLB, NKG2A, and TIGIT showed enhanced in vitro cytotoxicity against various AML cell lines [63]. Other groups have utilized CAR-NK92 cells to redirect the CD123 antigen, tested in AML cell-line-engrafted mice or patient-derived xenotransplantation AML models, and showed in vitro efficacy and systemic secretion of inflammatory cytokines, such as IFN-γ, but only modest AML control in vivo, mainly due to relatively short NK persistence and limited organ tropism [64]. CAR-NK92 cells have also been tested against mesothelin on the surface of AML cells demonstrating antitumor functionality [65]. The NK92 cell line provides a CAR-cellular platform that can easily be expanded and genetically manipulated; however, a cancerous cell line requires pre-infusion irradiation that can hinder its in vivo persistence and activity [11]. NK92 also lacks important activating NK cell receptors that can drive NK cell anticancer activity (such as CD16 and NKp44), compared to NK cells derived from peripheral blood [11].

Peripheral blood-derived NK cells (PB-NK) have been extensively studied as a source of CAR-NK cells for AML therapy because of their accessibility and ability to expand in vitro. PB-NK cells possess various activated receptors that render them mature, licensed, and highly cytotoxic [11]. CD33 and CD123 are antigens commonly targeted by PB-derived CAR-NK cells in AML. As discussed above, CD33 is highly expressed in AML and normal hematopoietic tissues [30]. Albinger et al., developed CD33 CAR-NK cells that exhibited enhanced in vitro cytotoxicity against an AML cell line and primary samples and increased leukemia control in an AML xenograft model after three infusions without any identifiable toxicities [66]. KO of the NKG2A inhibitory receptor in these cells further enhanced their anti-AML benefit [67]. Kararoudi et al., used CRISPR/Cas9 site-directed gene insertion to generate CD33.CAR-NK cells with different transmembrane and intracellular domains, showing antileukemic activity in vitro [68]. Garrison et al., developed the Senti-202, a CAR-NK cell product expressing simultaneously an activating CAR targeting CD33 and/or FLT3 on AML cells, an inhibitory anti-endomucin (EMCN) CAR that protects EMCN+ hematopoietic cells, and IL15 [69]. Senti-202 exhibited improved persistence and leukemic killing in vitro and in vivo as well as reduced toxicity against CD33+FTL3+ECMN+ hematopoietic stem and progenitor cells. 

Similar to CAR-T cells, CD123 has been investigated as a target in studies of CAR-NK cells in AML [70,71]. Our group tested a variety of CARs directed against CD123 with different intracellular combinations and found the 2B4.CD3ζ CAR to be the most optimal for CAR-NK activation against AML [71]. When further engineered with constitutively secreted IL15, CD123.CAR-NK cells exhibit increased anti-AML functionality and persistence in vivo. However, the IL15-secreting CD123.CAR-NK cells caused lethal systemic toxicity in AML xenografts, which was associated with the hyperproliferation of CAR-NK cells. Multiple infusions of CD123.CAR-NK cells without additional cytokines led to better AML control without systemic toxicity. As previously discussed, CD123 can also be normally expressed in bone marrow progenitor cells and endothelial cells, raising the concern for “on-target/off-tumor” toxicity. Carouso et al., compared the toxicity profiles of CD123.CAR-T and CD123.CAR-NK cells in mice engrafted with human hematopoietic cells and human endothelial tissues [72]. CD123.CAR-T cells were associated with significant myeloablation and increased infiltration and destruction of vessels, whereas their CAR-NK counterparts exhibited negligible endothelial toxicity and no ablation of hematopoiesis. Simultaneously, the CD123.CAR-NK cells were still capable of eliminating primary AML blasts in vitro and controlling the disease in AML xenograft models. Another study from Soldierer et al., revealed an optimal way of enriching CAR-NK cell isolates by using CD34 microbeads directed to the CD34 hinge on CAR-NK cells and subsequently evaluated both CD33- and CD123-based CARs in primary human NK cells, observing enhanced killing capacity against different AML cell lines and primary AML blasts in vitro [72]. While the addition of IL15 showed increased efficacy of CAR-NK cells in an ALL-xenograft model, no in vivo AML data were examined in this study. An additional approach to IL15 secretion is to stimulate CD123 expression. CAR-NK cell expansion is the rimiducid-based dimerization of inducible MyD88/CD40, which led to improved control of AML in vivo [73].

Recent advances in CAR-NK cell therapies have explored additional targets beyond CD123 and CD33 in preclinical models of anti-AML therapy. NK-cell-specific CARs have been developed with an extracellular moiety of the NK group 2D (NKG2D) receptor, an important NK-cell activation receptor that recognizes stress ligands on the surface of tumor cells and drives NK cell cytotoxicity and cytokine production [74]. Du et al., transfected PB-NK cells with genes encoding NKG2D CAR and hIL15. While NKG2D.CAR-NK cells demonstrated in vitro anti-AML cytotoxicity, the simultaneous ectopic production of hIL15 and three CAR-NK injections was necessary for sustaining the in vivo persistence and augmenting the in vivo anti-AML efficacy of NKG2D.CAR-NK cells [75]. The observed lack of IL15-mediated toxicity in this study may be due to the lower levels of secreted IL15. A comparative analysis of NKG2D CAR showed that CAR-T cells are more efficacious than CAR-NK cells against AML cells [76]. Another target is CD70, which, as mentioned above, is expressed in AML blasts and LSCs [77]. Choi et al., used CRISPR/Cas9 to KO CD70 in NK cells engineered with CD70.CAR and confirmed their in vitro efficacy against CD70+ tumor cells [78]. Gurney et al., followed a similar approach to target primary AML blasts in vitro using CAR-NK cells expressing an affinity-optimized CD38 CAR and further engineered with CD38 knockdown, given its expression in both AML and expanded NK cells [79,80]. CRISPR/Cas9 has mainly been used to KO the expression of target antigens in CAR-NK cells and abrogate the fratricide effect. However, Gurney et al., knocked out the checkpoint CISH gene from CLL-1-directed CAR-NK cells in an attempt to enhance CAR-NK anti-AML cytotoxicity, based on prior NK-cell studies that exhibited increased functionality on CISH KO NK cells [81]. Finally, cytokine-induced memory-like (CIML) NK cells, which are peripheral blood NK cells differentiated after brief pre-activation with IL12, IL15, and IL18, have been shown to exhibit antileukemic activity [82]. Dong et al., engineered CIML NK cells with a TCR-like CAR, recognizing a neoepitope derived from the cytosolic oncogenic nucleophosphmin-1 (NPM1c) presented by HLA-A2. These CAR-CIML NK cells can specifically target HLA-A2+/NPM1c+ AML cells in xenograft models [83].

CAR-NK cells generated by additional sources have been used in AML but are less extensive than the above-mentioned sources. Umbilical cord blood (UCB)-derived CAR-NK cells have been evaluated in a clinical trial of CD19+ malignancies, but only in vitro against CD123 and mesothelin on the surface of AML [84,85,86]. CAR-NK cells manufactured from induced pluripotent stem cells (iPSCs) have the feasibility of multiple genetic modifications during a very undifferentiated stage [11]. Two groups have engineered CD36KO iPSC-derived NKs with CARs (targeting NKG2D ligands or CD33), IL15 molecules, and a high-affinity, non-cleavable version of CD16 (hnCD16) capable of exerting antibody-dependent cell-mediated cytotoxicity (ADCC) mediated by a CD33 engager molecule or daratumumab [87,88]. All the preclinical studies of CAR-NK cells in AML are presented in Table 3.

Thus far, there are only limited data for the clinical efficacy of CAR-NK cells in AML. Only two phase I studies targeting CD33 in r/r AML have been completed, which showed safety but no objective responses (Table 4) [62,89]. The current clinical trials (active/recruiting) that utilize CAR-NK cells in AML are depicted in Table 5.

### 3.2. Advantages, Challenges, and Future Direction of CAR-NK Cell Therapy in AML

CAR-NK cells have multifaceted advantages over CAR-T cells in this context. One of the most significant advantages is the ability to use allogeneic sources of CAR-NK cells, which provides a potential “off-the-shelf” option for patients with aggressive neoplasms such as AML [91]. This approach has the added benefit of being readily available and does not require the time-consuming process of personalized manufacturing. Additionally, CAR-NK cells are inherently safer than CAR-T cells, as they do not carry the risk of graft-vs.-host disease, cytokine release syndrome, immune effector cell-associated neurotoxicity syndrome, or CAR-T cell-associated hemophagocytic lymphohistiocytosis [85]. Moreover, CAR-NK cells have a shorter lifespan than CAR-T cells, which can be advantageous in reducing the risk of “on-target/off-tumor” toxicity in tumors where the targeted antigens are also expressed in lower levels in normal tissues, such as AML [92]. Therefore, the use of CAR-NK cells in AML holds great promise and is an area of active research and clinical development.

On the other hand, despite their potential benefits, CAR-NK application in AML has several limitations. One major challenge is their short in vivo persistence, which makes the long-term immune surveillance difficult [71]. Immune surveillance is necessary for AML to prevent the relapse of residual LSCs, giving rise to a new generation of blasts. While additional modifications with cytokines can potentially address this issue, there is also a risk of inducing systemic toxicity, as we discussed previously [71]. It is possible that the most optimal strategy for the utilization of CAR-NK cells in AML is multiple infusions in a timeline individualized for every patient. Multiple infusions of CAR-NK cells require rigorous expansion and storage in cell banks. The in vitro expansion of CAR-NK cells to clinical-grade numbers is another challenge. Currently, NK cells mostly rely on feeder cells for expansion, which can limit their clinical applications [93]. Bead-based cytokine support has been developed and has the potential to solve this problem. Additionally, despite the relative plethora of preclinical studies of CAR-NKs in AML, only a few studies have been conducted at the bedside, which have shown overall unsatisfactory results in r/r AML, as described above. Apart from the reasons indicated earlier, this can also be attributed to the immunosuppressive effects on NK and CAR-NK cells. AML cells can alter the expression of ligands of NK-cell activation receptors, resulting in decreased targeting or the downmodulation of activating receptors [94,95,96]. Furthermore, AML can express molecules that trigger the downregulation of activating receptors and upregulation of ligands for inhibitory receptors or checkpoint molecules such as PD-1 and TIM3 [59,97,98,99]. Additionally, the hypoxic, glucose, and amino-acid-deprived tumor microenvironment (TME) of AML produces metabolites such as adenosine and suppressive cytokines such as IL10, IL6, and TGF-β, which all contribute to unfavorable metabolism and the suppression of CAR-NK cell activity [100]. To overcome these immunosuppressive effects, novel approaches are being investigated to target the metabolic regulation of CAR-NK cells (with cytokines or CISH deletion [101]), modulate the TME, or use combinations of CAR-NK cells with other immunomodulatory drugs or agents that reverse checkpoint inhibition. These strategies can enhance CAR-NK cell function and improve their therapeutic efficacy against AML.

## 4. Other CAR-Engineered Immune Effectors in AML

Apart from T lymphocytes and NK cells, other immune effectors have been evaluated for CAR-based strategies in AML. Cytokine-induced killer (CIK) cells are a special subtype of CD8+ T lymphocytes generated by the ex vivo incubation of peripheral blood mononuclear cells with IL2, anti-CD3, and IFNγ, which possess mixed characteristics of T and NK cells. Several studies have shown that CD33- and CD123-directed CARs engineered on CIK cells have efficacy against AML in vitro and in vivo, but with increased toxicity in hematopoietic tissues when targeting CD33 compared to CD123 [102,103,104,105,106]. To address this, Perriello et al., developed a dual CAR approach with a first-generation activating CD123.CAR and costimulatory CD33.CAR without activating domains to maintain anti-AML efficacy but circumventing anti-CD33-mediated toxicity [104]. Circosta et al., targeted the variant isoform 6 of the hyaluronic acid receptor CD44 (CD44v6), which is overexpressed in AML, and showed that CAR.CIK cells displayed enhanced leukemic killing in vitro against CD44v6 [107,108]. To increase the BM homing of CD33.CIK cells, Biondi et al., engineered them with CXCR4, achieving increased trafficking in BM [109]. Another lymphocytic effector is γδ T cells that contain TCRs comprising γ (gamma) and δ (delta) chains, which are MHC/HLA-independent and can be used in the allogeneic setting without causing GVHD. Delta One T cells (DOTs), an enriched γδT cell product, were modified with CD123.CAR and showed effective AML in vivo control after multiple cell infusions or a single infusion with daily IL15 support, with the latter condition exhibiting an anti-AML effect even after the tumor re-challenge [52].

## 5. Conclusions

In conclusion, although CAR-T cells targeting AML have not yet shown significant clinical responses, CAR-based immune therapy in AML is still evolving. The development of CAR-NK cells, which are considered safer and can be used “off-the-shelf”, has shown promise in preclinical studies, with ongoing efforts taking place for further optimization. For CAR-engineered immune cells in high-risk AML, it is important to consider the timing and setting of cell administration, as different engineered cells may be more suitable for different purposes, such as the facilitation of induction in refractory disease, bridge-to-HSCT, maintenance of remission, or remission after HSCT-relapse. Further preclinical research is needed to identify AML-specific antigens, optimize the CAR design with the incorporation of additional cell-specific co-stimulatory domains, investigate ways to overcome AML immunosuppression (combinational therapies with checkpoint molecules), reinforce the metabolic wellbeing of immune cells in the tumor microenvironment, and improve CAR-T and CAR-NK durability with a focus on memory subsets, among others. Additional clinical studies are also needed to identify the role of other CAR-immune cells in AML, improve the safety profiles and manage/mitigate the toxicities of CAR-T cells, and identify predictive biomarkers and refine patient selection criteria that can aid in patient stratification, predict treatment response, and guide therapy selection, ultimately leading to more personalized and effective CAR-treatment strategies in patients with AML.

## 6. Patents

IC has pending patent applications in the field of cellular immunotherapies.

## Figures and Tables

**Figure 1 cancers-15-03054-f001:**
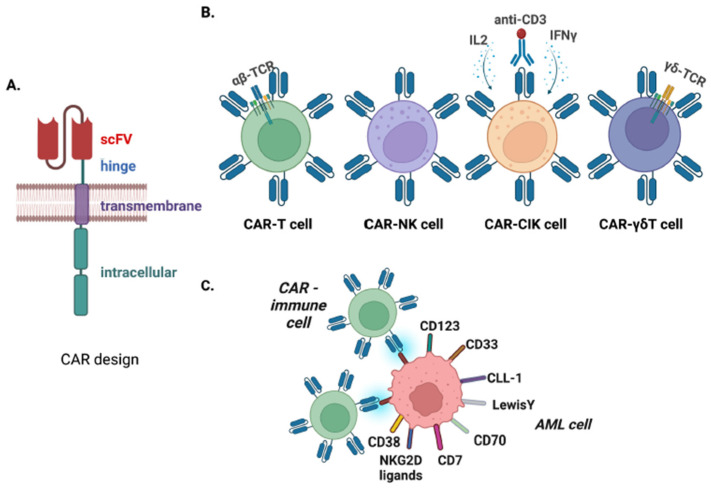
Chimeric Antigen Receptors (CARs) in Acute Myeloid Leukemia (AML). (**A**). Design of CARs with single-chain variable fragment (scFv), hinge, transmembrane, and intracellular domains. (**B**). Immune effectors that have been used in CAR-based immunotherapies of AML, including CAR-T cells (with αβ T-cell receptor/TCR), CAR-NK cells, CAR-CIK (cytokine-induced killer) cells, and CAR-γδT cells (with γδ TCR). (**C**). The most important antigens that have been utilized as CAR-target in AML. Figure was created using BioRender.com (accessed on 28 May 2023).

**Table 2 cancers-15-03054-t002:** Major limitations of CAR-T therapy in acute myeloid leukemia.

*Lack of AML-specific antigens* [46].
*Toxicities* (cytokine release syndrome, immune effector cell-associated neurotoxicity syndrome, “on-target/off-tumor” toxicities, CAR-associated hemophagocytic lymphohistiocytosis) [17,47].
*Need for autologous cell source* (*increased time and cost of manufacture for an aggressive neoplasm such as AML*) [49,50].
*Potential for targeted antigen downregulation from AML* [56].
*AML-induced immunosuppression* [46,57,58,59,60,61].

**Table 3 cancers-15-03054-t003:** Preclinical studies of CAR-NK cells in acute myeloid leukemia. PB: peripheral blood, CIML: cytokine-induced memory-like (CIML), iPSC: induced pluripotent stem cells, UCB: umbilical cord blood, KO: knock-out, KD: knock-down, hn: high-affinity, non-cleavable, FR: fusion-receptor, s: secretory, mb: membrane-bound, TriKE: tri-specific killer engager, N/A: information not available.

Targeted Antigen	Study	NK Source	Intracellular CAR Domains	Additional Genetic Modifications	Notes
CD33	[66,67] Albinger et al. [68] Kararoudi et al. [69] Garrison et al. [88] Wang et al.	PB PB PB iPSC	4-1BB-CD3ζ 4-1BB-CD3ζ and 2B4-CD3ζ N/A N/A	In [67] NKG2A KO FLT3-CAR, hIL15, EMCN hnCD16, mbIL15-FR, CD38KO	CRISPR/Cas9 FLT3 and CD33 CARs activating, EMCN CAR inhibitory Addition of daratumumab
CD123	[64] Morgan et al. [70] Kloss et al. [71] Christodoulou et al. [72] Carouso et al. [73] Wang et al. [86] Kerbauy et al.	NK-92 PB PB PB PB UCB	CD28-4-1BB-CD3ζ N/A 2B4-CD3ζ and 4-1BB-CD3ζ 4-1BB-CD3ζ -CD3ζ CD28-CD3ζ	sIL15 sIL15 sIL15, inducible MyD88/CD40, iCaspase 9 sIL15, iCaspase 9	Moreover, CARs with DAP10, FcεRΙγ Comparing CAR-T and CAR-NK
CLL-1	[77] Gurnley et al.	PB	N/A	CISH KO	CRISPR/Cas9
CD70	[78] Choi et al.	PB	N/A	sIL15, CD70 KO	CRISPR/Cas9
CD38	[80] Gurnley et al.	PB	CD28-CD3ζ	CD38KD	CRISPR/Cas9
NKG2D ligands	[75] Du et al. [76] Lelvas et al. [87] Davis et al.	PB PB iPSC	4-1BB-D3ζ 4-1BB-CD3ζ N/A	sIL15 hnCD16, mbIL15-FR, CD38KO	Comparing CAR-T and CAR-NK Addition of CD16-CD33 TriKE
mesothelin	[65] Tang et al.	NK-92	N/A		
NPM1c	[83] Dong et al.	CIML	4-1BB-CD3ζ	mbIL15-FR	

**Table 4 cancers-15-03054-t004:** Completed CAR-NK cell clinical studies with reported outcomes in relapsed/refractory acute myeloid leukemia. PD: progressive disease, CR: complete response, RD: relapsed disease, PB: peripheral blood, N/A: information not available.

Target Antigen	Phase	NK Cell Source	CAR Construct/Genetic Modifications	Reported Outcomes
CD33	I	NK92	[62] Tang et al.: N/A	3/3 PD.
	I	PB	[89] Huang et al.: N/A	4/10 PD, 4/10 RD after CR, 2/10 CR

**Table 5 cancers-15-03054-t005:** Active/recruiting CAR-NK cell trials in acute myeloid leukemia. r/r: relapsed/refractory, AML: acute myeloid leukemia, MDS: myelodysplastic syndrome, iPSCs: induced pluripotent stem cells.

NCT Identifier	Phase	NK Cell Source	Target Antigen	CAR Construct/Genetic Modifications	Disease	Location
NCT05574608	I	Unknown	CD123	Unknown CAR design	r/r AML	The 5th Medical Center of Chinese PLA General Hospital, Beijing, China
NCT05215015	I	Unknown	CD33, CLL-1	Unknown CAR design	r/r AML	Wuxi People’s Hospital, Wuxi, China
NCT04623944 [90]	I	Peripheral blood of haplo-matched, related, or unrelated donors	NKG2D ligands	NKG2D-OX40-CD3ζ Additional modification membrane-bound IL15	- r/r AML - intermediate, - high and very high-risk r/r MDS	Multicenter, Nkarta Inc., San Fransisco, CA, USA
NCT05247957	I	Cord blood	NKG2D ligands	Unknown CAR design	r/r AML	Hebei Yanda Lu Daopei Hospital, Langfang City, China
NCT05008575	I	Unknown	CD33	Unknown CAR design	r/r AML	Xinqiao Hospital of Chongqing, Chongqing, China
NCT05601466 and NCT05665075 [88]	I	Human iPSCs	CD33	Unknown CAR design Additional modifications - high-affinity, non-cleavable CD16 (hnCD16) - secretory IL-15 - CD38 knock-out	r/r AML	Institute of Hematology & Blood Diseases Hospital, Zhejiang University, Hangzhou, China
NCT05092451	I/II	Cord blood	CD70	Unknown CAR design Additional modification secretory IL15	- r/r B-Cell Lymphoma - r/r MDS - r/r AML	M.D. Anderson Cancer Center, Houston, TX, USA

## Data Availability

Not applicable.

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
