# Peer review of "A Panorama of Immune Fighters Armored with CARs in Acute Myeloid Leukemia"

_cancers, 2023, doi:10.3390/cancers15113054_

Round 1

Reviewer 1 Report

In this review, the clinical testing of CAR-T cells in AML and the limitations of CAR-T cell therapy were comprehensively summarized. Moreover, the clinical and preclinical landscape of CAR use in alternative immune cell platforms, with a specific focus on CAR-NKs were depicted. However, some details are still needed to be revised.

1.   The authors should supplement tables to depict clinical testing of CAR-T cells and preclinical and clinical studies of CAR-NKs in AML.

2.   Limitations of CAR-T cell therapy in AML should be divided into more detailed parts, and tables are recommended to be used to provide reference for future research.

3.   Further research and clinical trials should be elaborated to provide reference for researchers.

4.   The small font in Figure 1 affected the overall quality of the figure, further modification is required if possible.

5.   Was the author's name "Ilias Christodoulou1" miss spelled?

There are some spelling and grammar mistakes.

Reviewer 2 Report

Shoubld be published in cancers.

Author Response

Thank you for your kind suggestion

Reviewer 3 Report

In their paper, s Christodoulou and Solomou reviewed the available evidence on CAR-based therapy in acute myeloid leukemia.

The authors address the main issues of this therapeutic approach, well describing the current body of knowledge and the principal limitations of the different cellular therapy in the specific context of AML. The paper in comprehensive, well written in educational fashion and well balanced in terms of biological background and clinical data available so far. I have no major issues to raise.

Author Response

Thank you for your kind comments and suggestion 

Reviewer 4 Report

To the Author

In this manuscript, the importance of CAR-assisted immunotherapy in the treatment of AML is presented and evaluated using literature available to date.

The selected literature is chosen according to the claim of a review article and covers all major areas regarding CAR - based immunotherapy.

Suggestions for Improvement:

- In the context of quality assurance, the qualitative and quantitative detection of CAR-modified immune cells is of great importance for the progress of patients' treatment. The authors should evaluate and complement this issue with a short section.

Translated with www.DeepL.com/Translator (free version)

Author Response

"In the context of quality assurance, the qualitative and quantitative detection of CAR-modified immune cells is of great importance for the progress of patients' treatment. The authors should evaluate and complement this issue with a short section".

We appreciate your valuable comment regarding detecting CAR-modified immune cells in the context of patients' treatment progress. We have carefully considered your suggestion and have incorporated a short section within our manuscript at the end of section "2.1. Clinical testing of CAR-T cells in AML." This paragraph discusses the significance of qualitative and quantitative detection of CAR-T cells in monitoring treatment response and evaluating their efficacy. We emphasize the utility of flow cytometry and polymerase chain reaction (PCR) techniques for detecting CAR-T cells and assessing their persistence and expansion in patients. This short section aims to underscore the significance of qualitative and quantitative detection techniques in CAR-T cell therapy and their role in monitoring AML patients' treatment progress. We hope this addition provides readers with a broader understanding of the importance of quality assurance measures and the role of detection methods in optimizing CAR-T cell therapies in AML. We sincerely appreciate your insightful comment and guidance in enhancing the manuscript.